# Predicting Heavy Metal Concentrations in Shallow Aquifer Systems Based on Low-Cost Physiochemical Parameters Using Machine Learning Techniques

**DOI:** 10.3390/ijerph191912180

**Published:** 2022-09-26

**Authors:** Thi-Minh-Trang Huynh, Chuen-Fa Ni, Yu-Sheng Su, Vo-Chau-Ngan Nguyen, I-Hsien Lee, Chi-Ping Lin, Hoang-Hiep Nguyen

**Affiliations:** 1Graduate Institute of Applied Geology, National Central University, Taoyuan 32001, Taiwan; 2Center for Environmental Studies, National Central University, Taoyuan 32001, Taiwan; 3Department of Computer Science and Engineering, National Taiwan Ocean University, Keelung 202301, Taiwan; 4College of Environment and Natural Resources, Can Tho University, Can Tho 94000, Vietnam

**Keywords:** Random Forest, heavy metals, groundwater quality, explainable artificial intelligence (XAI), prediction intervals

## Abstract

Monitoring ex-situ water parameters, namely heavy metals, needs time and laboratory work for water sampling and analytical processes, which can retard the response to ongoing pollution events. Previous studies have successfully applied fast modeling techniques such as artificial intelligence algorithms to predict heavy metals. However, neither low-cost feature predictability nor explainability assessments have been considered in the modeling process. This study proposes a reliable and explainable framework to find an effective model and feature set to predict heavy metals in groundwater. The integrated assessment framework has four steps: model selection uncertainty, feature selection uncertainty, predictive uncertainty, and model interpretability. The results show that Random Forest is the most suitable model, and quick-measure parameters can be used as predictors for arsenic (As), iron (Fe), and manganese (Mn). Although the model performance is auspicious, it likely produces significant uncertainties. The findings also demonstrate that arsenic is related to nutrients and spatial distribution, while Fe and Mn are affected by spatial distribution and salinity. Some limitations and suggestions are also discussed to improve the prediction accuracy and interpretability.

## 1. Introduction

Water quality sampling is essential to reflect the environmental status and governance efficiency. However, spare monitoring data and tedious analysis processes also inhibit the interpretation of ongoing pollution events, leading to broader contaminant transportation and treatment costs [1]. Heavy metals are not only among the toxic substances causing a high risk to ecological and human health but also need much time and complicated procedures to detect and remediate. Thus, developing a model to predict heavy metals will optimize the data collection process, transmission, and lab-based analyses [2,3], resulting in a faster response to contamination. Current technology advancements enable real-time water quality monitoring, but continuous monitoring at dense spatial scales will be a cost challenge. Most sensor technologies are expensive and need regular care but may measure a single or a few parameters [4]. Therefore, utilizing data that can be measured from one in situ multiparametric probe to predict heavy metals would be very applicable for setting up an effective monitoring system.

The practical approach to evaluating a model’s feasibility relies on the performance of the model, including the data accessibility and the model’s robustness, accuracy, and uncertainty. The feasible model could support the stakeholders in deciding whether the target task is successful and applicable. Many studies have successfully applied cost-effective parameters and machine learning algorithms for predicting water quality indexes [5,6] or heavy metals [7,8]. However, the previous investigations usually focused on accessing model predictability for one data set without uncertainty evaluation. Evaluating a trained model based on historical data might be too optimistic because future data behavior could change with different degrees of uncertainty induced by natural or anthropogenic activities [9]. Therefore, comparing single values of evaluation metrics, such as goodness-of-fit and errors, could not reveal much about the prediction perturbation [10]. Modelers need more evidence beyond accuracy to convince the general public or managers to trust predicted values [11]. Thus, efforts need to be devoted to exploring how the changes in data input may lead to a different prediction, whether the cost-effective features can yield good predictions, and which factors drive the outcomes.

Furthermore, from a quick overview, studies that express interpretability in heavy metal prediction models have not yet been developed and assessed. The wisely used analyses, namely the correlation analysis, clustering analysis, and principal component analysis, have been applied to explore the causal relationship between groundwater heavy metals and other chemical compositions [12,13,14,15,16,17]. Those outcomes could explain the contamination sources or geochemistry conditions but could not provide information on how a predictive model could internally make decisions. The interpretable or transparent models with clear links between input and output are often insufficient to express highly non-linear relationships [18]. Although the black-box or opaque models are more accurate and reliable, operators and managers have found it difficult to trust and implement them [19]. Even transparent models occasionally require explanations for non-expert audiences [20]. A model with explainability will aid in expressing their results as convincing evidence for their real-life implementation, improvement, and transmission.

The study aimed to develop a scanning framework to test the feasibility and interpretability of a generalized machine learning model for heavy metal prediction. With the recent development of sensor technologies, low-cost multiparameter probes are widely used for measuring fundamental groundwater quality. However, determining various heavy metals remain challenging because of the cost–demand of field samplings and the associated laboratory work. The observations of multiparameter probes could facilitate real-time predictions of heavy metals. In the study, this modeling experiment focused on the feasibility of available physicochemical parameters by comparing a model’s performance on data perturbation, feature changes, and prediction intervals. The proposed model can be reliable and accessible for practical applications if the low-cost parameters outcompete other predictors. Furthermore, the model interpretability is the supplementary information to enhance a user’s understanding of the rationale behind the prediction or how much each predictor contributes to the estimated value. We expect the systematic assessment to provide general insight into the modeling issues of accuracy, reliability, and explainability to facilitate the trained models in groundwater quality monitoring and management.

## 2. Materials and Methods

### 2.1. Uncertainty Quantification

Several approaches have been proposed to quantify uncertainty, including the Monte Carlo methods [21,22,23,24], perturbation methods [25,26,27,28], or Bayesian methods [29,30,31]. In particular, most approaches can only measure specific uncertainty sources. The Monte Carlo methods focus on parameter uncertainty, Bayesian methods deal with input data uncertainty, and perturbation methods optimize model architecture robustness. These approaches require the known parameter distributions, a constraint in sparse sampling datasets. Nonetheless, bootstrapping does not need either assumptions about the data distribution or complex computations [32]. Bootstrapping is compatible with assessing the uncertainty of unknown or complicated data distribution and the insufficient sample size of a large-scale model. In practice, the bootstrap method has been applied in model robustness evaluation by comparing the distribution of model performances [10,33] and prediction interval estimation [34,35,36,37].

In this computational experiment, we considered uncertainty in various sample distributions. The bootstrap scheme was set up 100 times, randomly splitting 75% of data for training and 25% for testing using different random states. One hundred times replicated sampling with replacement from the data pool will give 100 prediction results. Consequently, repeated in-samples (training data) were used for training models, whereas we only used out-samples (testing data) for uncertainty evaluation. The distribution of 100 evaluation metrics in the box and whisker plots can demonstrate the error range that each model produces. Similarly, the model’s performance distribution on feature selection uncertainty was assessed on different feature groups of 100 bootstrap datasets. Finally, the models or predictors with the lowest bias range and highest goodness-of-fit were selected for the subsequent analysis.

### 2.2. Interpretable Machine Learning 

Explainable artificial intelligence (XAI) relates to using methods that explain how machine learning algorithms make decisions so that humans can understand them [38,39,40,41]. Concerning the scale of interpretability, global explanations determine which variables have the most predictive power, while local interpretations estimate how much each variable contributes to a prediction [42]. Permutation Feature Importance (PFI) is a global and model agnostic method that can be used for any model in selecting the most relevant features for predicted targets to decrease the required data and computational cost [43,44,45,46]. PFI only shows how sensitive each variable is but does not show how negatively or positively those features contribute to the model’s output [47]. To obtain local explanations of a single prediction, many studies [48,49,50,51,52] have applied local interpretable model agnostic explanations (LIME) and Shapley additive Explanations (SHAP) because of their attractive visualization and multi-properties. However, SHAP-Kernel and SHAP-Deep Explainer are very slow and do not support some models. Feature coefficients provided by LIME are similar to the coefficients extracted from Bayesian linear regression and feature importance from decision trees [52]. For tree-based models, Treeinterpreter [53] outperformed SHAP-Tree Explainer [50] in terms of attribution accuracy computation cost [54].

### 2.3. Conceptual Framework

In this study, many popular machine learning algorithms such as Support Vector Regression (SVR), K-Nearest Neighbors (KNN), Feedforward Artificial Neural Network—Multilayer Perceptrons (MLP), Random Forest Regression (RFR), Gradient Boosting Regression (GBR) and Linear Regression (LR) are applied to achieve the objectives. Firstly, data discovery and pattern mining methods included clustering data by Density-based Spatial Clustering of Applications with Noise (DBSCAN) to find the similarity samples and outliers; we also analyzed multivariate statistics to discover the relationship between hydrochemical parameters. Secondly, the baseline model was trained with the most abundant cluster data, which was derived from DBSCAN, using all physicochemical parameters (Depth, Temp, EC, pH, TH, Cl, TDS, NO3, NH4, SO4, TOC) and two spatial features (X, Y geographic coordinate). We used a performance metric range from lower and upper bounds to express prediction uncertainty instead of a single value. A bootstrapping sampling was applied to generate sample variability for input perturbation. The distribution of model performance was evaluated to select which was the most suitable model. The best-performed model was selected for further analysis. Besides that, the selected model was optimized by a five-fold cross-validation of training data and the RandomsearchCV method to find the number of optimal features for each target. The tuned hyperparameters included max_depth, min_samples_leaf, max_features, min_samples_split, and n_estimators. After optimizing, the models were applied to assess the feature contribution by ranking in PFI. Thirdly, we examined the feasibility of low-cost parameters for prediction by comparing the performance distribution between three feature combinations: (1) the first feature set consisted of full parameters; (2) the second feature set was selected by an optimal number of features that have a high rank in PFI; (3) the third feature set followed the criteria of technical and economic efficiency that are quick and inexpensive to measure on-site, irrespective of its sensitivity. Finally, model uncertainty was quantified by the model prediction intervals (PI) and the percentage of expected values falling into the intervals. The added interpretability analysis also enhanced the model’s trustworthiness. The general research procedure is illustrated in Figure 1. Although the applied methods in the framework are not highly complex, they are easy to use and computationally efficient. With the combination of all the procedures, we were able to quickly discover new insights into the capacity of the indicators and models.

The study was conducted using Python scripting on the PyCharm integrated development environment. Scikit-learn [55], Matplotlib [56], Pandas [57], Numpy, Treeinterpreter, and other packages were used to perform the analysis, simulation, and output charts.

### 2.4. Evaluation Metrics

There were three metrics used for the model’s evaluation. The R^2^ (the coefficient of determination) is a typical metric for quantifying the variance in outputs of linear models [58], indicating the goodness of fit (Equation (1)). The R^2^ score ranges from −∞ to 1; thus, the closer to 1, the better the prediction is. Root mean squared error (RMSE) is the standard deviation of the residuals, computed as Equation (2). When compared to mean absolute error, RMSE can provide a more reliable error distribution and sensitivity within a large sample size [59]. Moreover, the model’s uncertainty is expressed by the Prediction Interval Coverage Probability (PICP) to find the possibility of expected values falling into the interval (Equation (3)) and the Mean Prediction Interval (MPI) (Equation (4)) to indicate the average width of all prediction intervals. The lower the MPI, the lower the uncertainties. If PICP is greater than the probability quantile, the uncertainty is overestimated, otherwise it is underestimated. PICP and MPI have been used in [22,60,61,62] to evaluate prediction uncertainties.
(1)$$R^{2}(y,\hat{y})=1-\frac{\sum_{i=1}^{n}{\left(y_{i}-{\hat{y}}_{i}\right)}^{2}}{\sum_{i=1}^{n}{\left(y_{i}-\bar{y}\right)}^{2}}$$
(2)$$RMSE=\sqrt{\frac{1}{n}\sum_{i=1}^{n}{\left(y_{i}-{\hat{y}}_{i}\right)}^{2}}$$
(3)$$PICP=\frac{count(N|Qlow\leq N\leq Qhigh)}{n}*100\%$$
(4)$$MPI=\frac{1}{n}\sum_{i=1}^{n}\left(Qhigh-Qlow\right)$$
where *y_i_* is the observed values, and *ŷ_i_* is the estimated values. Notations *Qlow* represent lower quantile, and *Qhigh* is for the upper quantile.

### 2.5. Study Site and Data Sources

Taiwan covers 35,808 square kilometers (35,801 km^2^), with 70% coverage of rugged and densely forested mountains as the spine in the central. The flatter area is located along the west coast, which is also densely inhabited. Uneven rainfall and steep-sloped rivers force residents to depend on groundwater resources significantly [63]. Since 2002, the Taiwan Environmental Protection Administration (EPA) has monitored groundwater with a relevant sampling frequency (monthly or seasonally, half-yearly) at different authority levels. All the monitoring data are updated on the EPA Taiwan website for public access, whether being single-well-, administrative- or watershed-level [64]. The study obtained physicochemical data of groundwater monitoring data from 453 wells over ten groundwater basins of Taiwan from 2000 to 2020. The location of monitored wells and the defined basins are shown in Figure 2.

In this study, the water quality records were downloaded from the Taiwan EPA website [65]. According to the Taiwan Groundwater Pollution Monitoring Standard, As and Mn concentrations for domestic supply and irrigation purposes are limited at 0.025 mg/L and 0.25 mg/L, respectively, whereas Fe values are 0.15 mg/L and 1.5 mg/L, respectively. The 2019 annual water quality monitoring report of Taiwan EPA revealed that only 53.2% of manganese (Mn) samples achieved the standard, while 73.4% of iron (Fe) is under-controlled [64]. A quick scan of collected data also shows that Mn, Fe, and As comprise levels of 27%, 10%, and 0.3% in samples, respectively, exceeding the standard levels, while other trace elements (Cr, Cd, Cu, Pb, and Zn) are much lower than the limits. Therefore, As, Fe, and Mn were targeted in this experiment. Fourteen physicochemical constituents were used for analysis. After the data preprocessing, there were 20,685 groundwater samples in the period of 2000–2020 used for modeling. Table 1 shows the statistical summary of the data.

## 3. Data Analysis and Feature Engineering

The clustering analysis by DBSCAN shows the distribution of groundwater quality deviating through a spatiotemporal scale. The algorithm generated six unique clusters, including five clusters (from 0 to 4), based on their similarities and noise data (cluster-1). To illustrate how the clusters distribute across spatial scales, we added geographical coordinates and show them on the scatter chart in Figure 3. The concentration of target parameters (Mn, Fe, As) from each cluster is shown in Figure 4. There is a distinguishable characteristic of different clusters. For instance, Cluster 1 contains all extreme values considered outliers from most of the sampling locations. Cluster 0 can represent groundwater quality in Taiwan because it includes the most abundant samples in over ten groundwater basins. On the other hand, Cluster 1 to 4 have a smaller sample size, characterized by small As, Fe, and Mn concentrations in specific basins. Concerning temporal distribution, Cluster 0 is similarly distributed throughout the years, but minor clusters are scattered/distributed in some periods. In order to prepare data for As, Mn, and Fe prediction, Cluster 0 was used for training and testing the datasets, while all the other clusters with particular characteristics were filtered out.

The Spearman correlation coefficient was applied to assess the interrelationship between parameters in Cluster 0 because this method can capture the non-linear relationship between variables. The coefficients are close to 1 for a similar trend and −1 for an opposite trend, while nearly 0 means no relationship. Figure 5 shows a highlighted relationship between salinity indicators, namely EC, TDS, Cl, SO4, and TH. Moreover, salinity indicators have a slight opposite trend with X. It can be interpreted that the larger X values (from the west coast toward the east), the lower the salinity levels are. There was no clear relationship between Y and salinity. The water temperature slightly increases toward the south (decreasing Y) and the west (decreasing X). Considering the feature–target relationship, As, Mn, and Fe have a slight negative correlation with NO3 and a positive correlation with NH4. The opposite trends of NO3 and NH4 can result from the oxygen concentration in water. Saltwater reduces dissolved oxygen, leading to an anoxic condition and denitrification. Therefore, As, Mn, and Fe may be related to the salinity level. Salinity had robust effects on Mn and Fe solubility [66]. Other features do not have a clear correlation with targets, but they may have non-linear relationships.

Preprocessing steps included filling missing values, removing bad attributes, adding features, resampling, and normalizing samples. In consideration of the spatial heterogeneity of samples, there is no other convenient information available except for locations. That is why longitude and latitude coordinates (X, Y) were also used as spatial feature inputs for prediction. To reduce the model complexity for point estimation, we shuffled the samples before the train–test partition and ignored the temporal dimension. Moreover, the sample size at each location should be over 30 samples for meaningful statistical analysis. As a result, from 403 wells of Cluster 0, we have 20,685 samples for training and testing. Finally, all input features were normalized into a fixed range between 0 and 1 to speed up the computation.

## 4. Results

### 4.1. Assessment of the model Predictability

It is necessary to evaluate model robustness by comparing result distribution since average metrics cannot precisely represent how stable the model performance is against new inputs. Initially, we trained six regression algorithms (MLP, RFR, KNR, SVR, LR, and GBR) to predict As, Fe, and Mn, using all features for 100 data sets to find which model is the most robust. Because R^2^ scores can have negative values, the RMSE values were used for plotting charts. The prediction biases of As, Fe, and Mn from 100 random datasets are shown in the box and whiskers plots in Figure 6 to illustrate the stochastic nature of the algorithm’s performance due to input changes.

Overall, the RMSE distribution of RFR in all target species (Figure 6a–c) has the lowest mean errors in the training and testing data sets, indicating that RFR produces fewer biases than other models. From As prediction (Figure 6b), all models (except for SVR) are robust to data perturbations because their interquartile ranges of errors are small and less susceptible to outliers. There are also tiny gaps between training and testing performances caused by the similar training and testing data distribution or generalized models. However, Fe prediction (Figure 6b) and Mn prediction (Figure 6c) experienced similar problems, with potential outliers resulting from noisy data sets or highly skewed distributions. The significant differences in the training and testing error distributions of RFR and KNR exhibit overfitting models. In fact, it is hard to build a model that is ideally fitted to all new data. Overfitting models learn noise rather than actual signals, but they can be improved through hyperparameter optimization or input regularization. That is why RFRs were considered even though they are overfitted. Those models with a low variance and high bias may need more data or features that we did not consider in this scope.

In addition, the testing R^2^ scores of the RFR models for all targets are the highest, followed by KNR and GBR (Table 2). All RFR models predicting As, Fe, and Mn have high average fitting scores (over 0.7), while LR and SVR seem unsuitable for the data. LR and SVR have average R^2^ scores lower than the satisfactory criteria (<0.5). Therefore, RFR was selected to predict As, Fe, and Mn.

### 4.2. Assessment of the Feature Predictability

The predictability of feature sets was evaluated by comparing performance metrics between feature groups toward each target. Then, the feature sets were selected through three steps.

Firstly, three RFR models for three targets were successively optimized by Random search CV and five-fold cross-validation to find the optimal values for hyperparameters. Some primary hyperparameters were used for tuning, namely max_depth, min_samples_leaf, max_features, min_samples_split, and n_estimators. The best configuration of each model is reported in Table 3. Figure 7 shows the validation curves for the number of features from each model. The red lines from Figure 7a–c are the optimized feature for As, Fe, and Mn models. For instance, the As model needs a maximum of ten features, while Fe and Mn models require six and eight features, respectively. Although the gaps between train and test errors can remain somewhat large at the stopping points, the validation scores will not decrease due to overfitting. An efficient model will require less input but perform satisfactorily for testing data. Fewer redundant features mean fewer opportunities to make decisions based on noise. Therefore, fewer features reduce the algorithm’s complexity, resulting in faster model training [67].

The learning capability of the optimized models is expressed by learning curves, which show the behaviors of training scores and validation scores responding to sample numbers. From Figure 8, the training and validating curves have not yet converged as the data are increasing. Variability during cross-validation was higher than during training, showing that the models suffer from variance (overfitting) rather than bias. The dataset is so unrepresentative that the model cannot capture the statistical characteristics. Potentially, the validation scores could increase and would be closer to the training scores if more data are trained to generalize more effectively.

Secondly, as to which variable is selected for each group depends on the order in the feature importance process. Figure 9 reveals the sensitivity of each feature to the predicted targets. It can be seen that NH4, NO3, pH, X, and Y showed a strong effect in the three models, while TOC and Temp had a negligible impact on the results. As expected, the correlation of NH4 and NO3 with targets also affects the importance of analysis. pH, X, and Y have unclear correlations with targets but also have high importance. This result also proves that the most important features are not always the most statistically correlated features [46]. The ionic forms of heavy metals also relate to pH levels and oxidation–reduction conditions. For instance, increasing pH values cause the heavy metal precipitation to increase, whereas the solubility decreases [68,69,70]. The high ranks of X and Y show that spatial heterogeneity rather than physicochemical parameters influence the results. Additionally, RFR often gives low importance to those features with collinear relations, such as EC, TDS, although they may have physical meaning to the targets.

Based on the optimized number of features required for each model in Figure 7 and feature ranks in Figure 9, we divided the input features into three groups: full features, important features, and low-cost features, as given in Table 4. Essential feature sets were selected from the highest downward until the required max features. The low-cost features should be quick and inexpensive to measure by one multiparameter probe; thus, a slow or expensive feature from the important feature sets was replaced. For instance, SO4 is harder to measure by the same probe; it was replaced by EC (in the As model) or Cl (in the Mn model) as a low-cost feature. Finally, although important features from the Fe model can be measured easily by one multiparameter sensor, another low-cost feature set was created to evaluate.

Finally, we compared the performance distributions of the RFR between full features and reduced inputs on 100 testing data sets. This experiment aimed to find the most robust features for each model. The goodness-of-fit distribution was compared in Figure 10. In general, all feature sets successfully fit targets with a minimum R^2^ score higher than 0.6. The As model performances by three feature groups were almost similar (Figure 10a), while the reduced feature sets of the Fe and Mn model caused a decrease in model performance (Figure 10b,c). The Wilcoxon signed-rank test was applied to calculate the performance discrimination among benchmark models (full features) and other feature sets. The null hypothesis was that the performance of the paired feature sets is similar. If the *p*-value is more significant than 0.05, they have similar distributions; otherwise, they come from different distributions. As shown in Table 5, the important feature scores and the low-cost feature scores of As model are similar to the full feature scores; thus, using low-cost features is more beneficial. Although the Fe model’s important feature scores and low-cost feature scores are different from the full feature scores, the average low-cost feature score is higher and more robust to data changes in the Fe model. In Mn prediction, both important feature scores and low-cost feature scores have the same distribution and are worse than the full feature scores. Therefore, low-cost features can predict As, Fe and Mn.

### 4.3. Quantification of Predictive Uncertainty

This evaluation explains how confident the models can perform on one data set. The prediction interval for a response is constructed from the results of single decision trees in the optimized RFR. In order words, each output value from the As model, Fe model, and Mn model was aggregated from 1848, 1727, and 1000 possibilities, respectively. We expected the PICP of 90% prediction interval to be around 90%. The lower values of MPI are the lower uncertainties. Table 6 summarizes the comparison of prediction uncertainties generated by different feature sets. The full features produce the highest uncertainties among the three feature sets, whereas the low-cost features generate almost the lowest uncertainties. Very high PICPs of the As model by three feature sets indicate that the model overestimates the noise, while the Fe and Mn models underestimate the uncertainties. The real uncertainties may be higher than calculated, but the Mn models are more reliable than the Fe models. Hence, the low-cost features can predict heavy metals but need more regularization to improve the model’s generalization.

Uncertainty simulation results of low-cost features for three models are illustrated in Figure 11. Overall, the observed values tend to fall into the prediction intervals, masked by the red shading areas. The median predictive performance of the As model (R^2^ = 0.80, RMSE = 0.006) is the highest, followed by the Mn model (R^2^ = 0.75, RMSE = 0.401) and Fe model (R^2^ = 0.65, RMSE = 2.219). It can be seen that the Fe model and Mn model cannot capture data patterns well, resulting in high variance. The reason may come from the lack of good predictors because the absence of causal links will limit machine learning algorithms from drawing desirable outcomes [11]. Besides, bias could be yielded from too few features or inference of false feature relationships [19]. Reducing features will ignore some essential features relevant to predictability and cause a significant impact on predictability. This is contrary to the Fe model. Even the full features also cause the largest uncertainties. This situation may result from unrepresentative data.

### 4.4. Interpretability of the Proposed Models

This analysis was aggregated from Treeinterpreter outputs. The predicted values were decomposed into a linear combination of feature contributions and biases to understand how the model estimates those targets. The bias was assumed to be the same for training and testing data; thus, the differences in feature contributions will produce variability. Figure 12 shows the coefficient distribution for each target in training and testing data. These graphs describe the local interpretability for a single prediction. NO3 and NH4 have the most significant contribution range, while the other features are susceptible to outliers, indicating scattered or non-normal distributed data.

In general, the global feature contributions were calculated by averaging the local coefficients of each prediction (Table 7). Nutrients, spatial characteristics, and salinity highly contribute to As, Fe, and Mn predictions. Under more reduction conditions, Fe and Mn have more extensive concentration ranges [71]. These results are suitable with earlier correlation analyses and explanations. The dissimilar ranking in feature contributions in training and testing shows that either the decomposition method cannot explain the Fe and Mn models or that the training and testing sets come from different distributions. Moreover, the feature contribution ranking presented by the three models in the training stage is different from the ranking in feature importance (see Figure 9), indicating that the models do not generalize well [72]. In order to have a more efficient explanation of model behaviors, samples with different characteristics can be separated into different explanatory models or even different predictive models. Due to time and computation constraints, the interrelationship of features has not been calculated.

## 5. Discussion

Although low-cost features can potentially predict heavy metals through this evaluation framework, both model configuration and feature sets are not fine-tuned for their actual application. Therefore, some issues need to be considered to yield better results.

Firstly, the selected models are not finely optimized due to the shortcomings of the randomized search method, namely, not fully exploring the hyperparameter search space. That is why the models are overfitted. They can be improved by applying other exhaustive optimization methods, such as Bayesian optimization or Grid search. However, Random Search could be relatively close to optimal performance but requires less computation for large sample sizes and many hyperparameters [73,74]. It is still suitable for quick scanning over reasonable hyperparameter ranges.

Secondly, the feature selection method by importance rank may have an adverse effect. It gives a single impact on model performance, whereas the inter-relationships are hidden. Hence, the selected features cannot capture data characteristics. Even though we used all the feature sets, the training scores of Fe and Mn were not too high. In both the As and Fe models, uncertainties were unavoidable due to the inherent uncertainties of the hydrological process [75]. Koutsoyiannis, 2003 [76] also found that hydroclimatic processes produce more uncertainties than estimation because assumptions are often based on a stable climate. In order to understand the multi-scalar behavior of contaminated substances at multiple spatio-temporal scales, using the entropy technique and the Hurst exponent can be very helpful [77]. However, natural structural factors, including climate, soil and aquifer formation, sources of groundwater recharge, or human impacts, may interrupt the long-range correlation of hydroclimatic processes [78]. Vu et al., 2019 [79] and 2021 [80] also found that land use, especially in agriculture-dominated regions, has a higher potential for groundwater contamination; thus, land use can be a good indicator of groundwater characteristics. A simple set of predictors cannot perform as well as expected; thus, adding more features may improve the model’s performance.

Moreover, the learning curves indicate that adding more samples or modifying the data structure may improve the model performance. A suitable dataset is more effective than many predictors and sample sizes [81]. Data regularization can also smooth the noise to get more generalized predictions. Due to training the models on bootstrapping samples, the Hurst phenomenon in hydrological processes was excluded. Analyzing Hurst–Kolmogorov dynamics to find independence structures [82] and cluster data in smaller groups and training by different models is another strategy to cope with data heterogeneity at different scales. Simplifying the models then necessitates further effort to explore data distribution, instance structures, and target analysis to enhance the scalability of the predictive models. However, training data subsets with different models will amplify computational load and management. Improving learning efficiency can be done by simultaneously computing services [83].

Lastly, uncertainty quantification and model interpretation show the heterogeneity in the data structure. By using gene entropy techniques and the Hurst exponent to understand the multi-scalar behavior of nitrate-N in groundwater, this approach should readily be transferable to other contaminated aquifers and catchments. A realized model should increase accuracy by increasing the complexity, such as adding multi-source data collection and internet of things (IoT), geographic information systems (GIS), and numerical models into a large-scale model [84] to have more precise simulation at each spatial scale. As a result, the model may lose interpretability. The model interpretation can also be processed before optimization to remove unnecessary features. The selection of quick scan techniques should be prioritized to save time and effort. Therefore, seeking a good model and indicators is an exhaustive trial–error process.

## 6. Conclusions

The study has developed a scanning framework to test the feasibility of candidate machine learning models, including Support Vector Regression (SVR), K—Nearest Neighbors (KNN), Feedforward Artificial Neural Network—Multilayer Perceptrons (MLP), Random Forest Regression (RFR), Gradient Boosting Regression (GBR) and Linear Regression (LR) and interpretability of the selected model. The samplings and experiments for heavy metals in groundwater have been challenging tasks and have drawn much attention in environmental sciences. The main issues are the test costs that grow exponentially with the concentration accuracy of the target heavy metals. The prediction of heavy metals using low-cost water quality data can considerably benefit the monitoring efficiency of heavy metals in groundwater systems. The study evaluated the predictability of machine learning models, compared the performance of different water quality indicators, and explored the reliability and explainability of the model for practical applications.

This study observed that heavy metals in groundwater, namely As, Fe, and Mn, could be predicted by Random Forest using low-cost physicochemical water quality samples. Results showed that the As model using ten predictors (NH4, X, Y, pH, NO3, EC, Cl, Depth, TH, TDS) had low bias (R^2^ = 0.80) and low uncertainties (PICP = 97.97%, MPI = 0.015). The Mn model using eight features (pH, Y, NO3, TH, NH4, Cl, X, EC) yielded a relatively satisfactory fit (R^2^ = 0.75) and slightly high uncertainties (PICP = 86.41%, MPI = 1.230). The Fe model needed the fewest features (pH, NH4, Y, EC, Depth, X) but has not been generalized, leading to the lowest fitting scores (R^2^ = 0.65) and high uncertainties (PICP = 77.13%, MPI = 4.907). The feature contribution simulations showed that nutrients, salinity, and spatial factors strongly affect heavy metal behaviors.

In the study, the predicted values were decomposed into a linear combination of feature contributions and biases to understand how the model estimates those targets. Results showed that NO3 and NH4 had the most significant contribution range, while the other features were susceptible to outliers, indicating scattered or non-normal distributed data. Nutrients, spatial characteristics, and salinity highly contributed to As, Fe, and Mn predictions. Although the built models were not optimally generalized, the results were promising. This framework addressed how accurate and sensitive the model performances were, how confidently the predictions covered, where the uncertainty sources were coming from, and how a particular instance was predicted. The limitation of this experiment is that one generalized model cannot fit all data patterns in a large area. Further investigation for more features, namely vegetation cover, climate, topography, land deformation, or soil properties, may improve the results. Because the data are heterogeneous, it can be localized in smaller groups by time, space, or chemical characteristics (hardness, salinity) before training. Other techniques, like clustering, classification, and mixture models, can be combined to improve the regression performance. However, those findings help in assessing the practicability of the proposed model for groundwater quality inspection.

## Figures and Tables

**Figure 1 ijerph-19-12180-f001:**
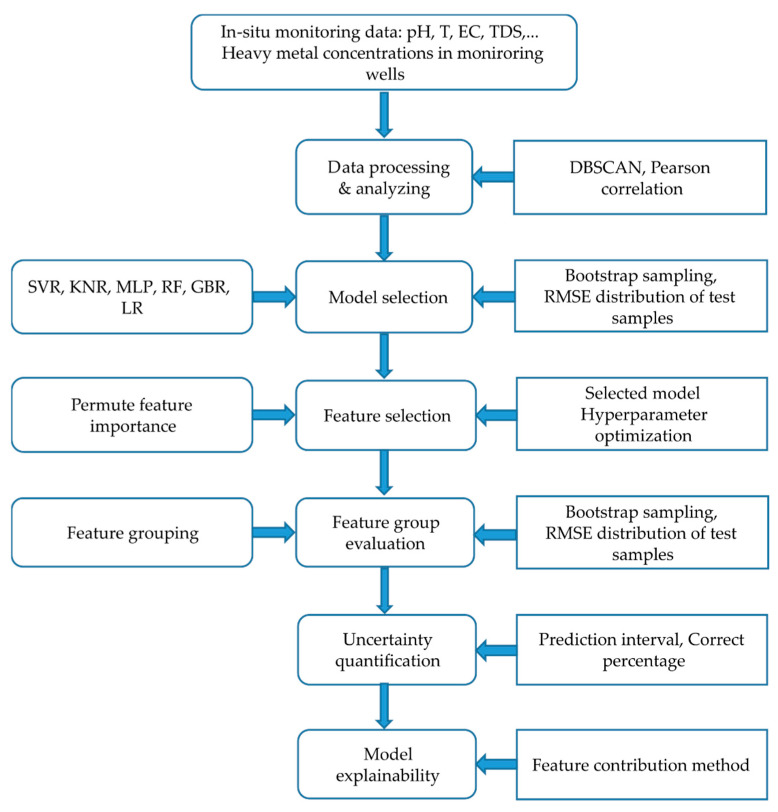
A Conceptual research framework.

**Figure 2 ijerph-19-12180-f002:**
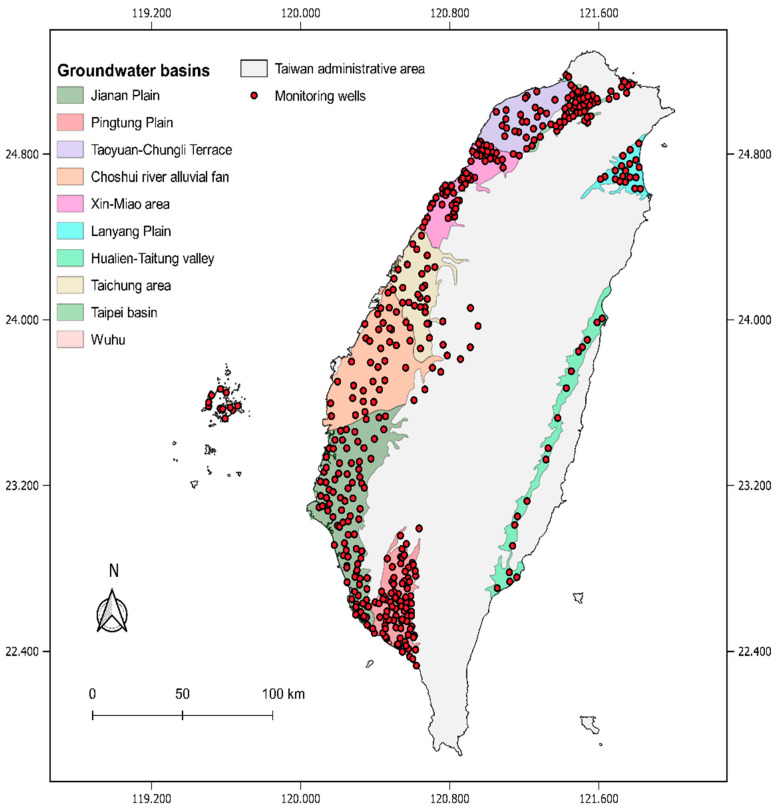
The locations of 453 observation wells in ten groundwater basins in Taiwan.

**Figure 3 ijerph-19-12180-f003:**
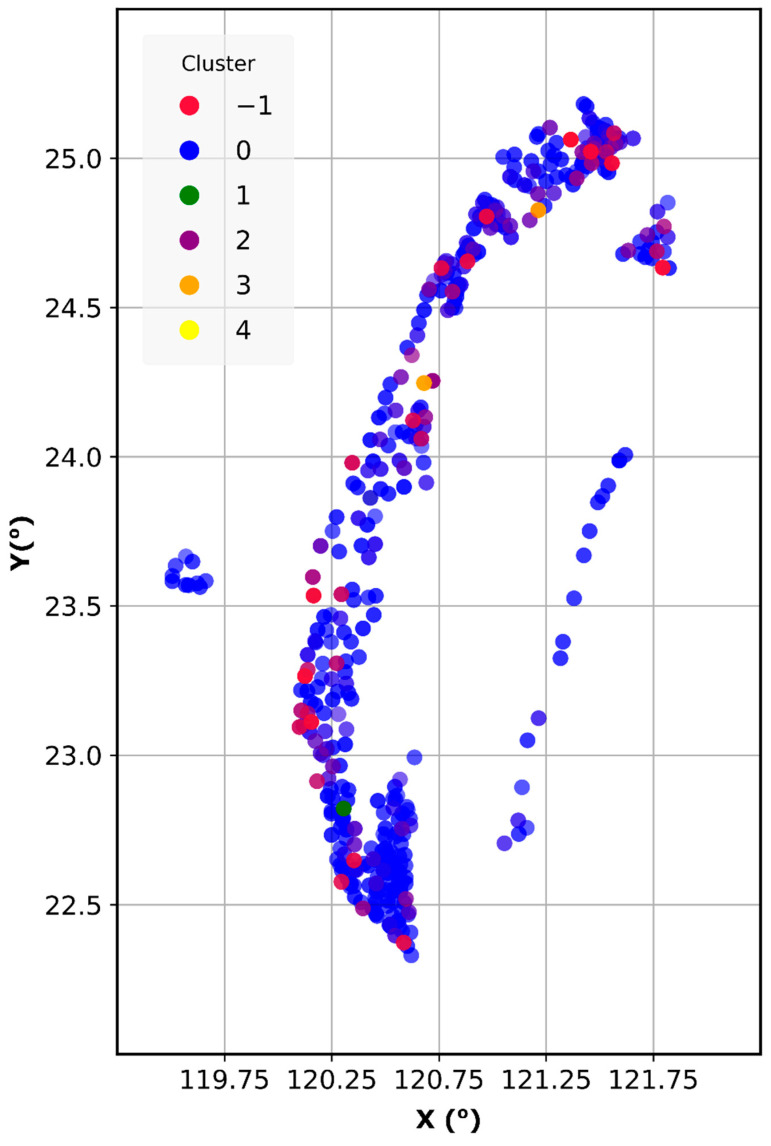
Scatter plot showing the spatial distribution of derived clusters and outliers.

**Figure 4 ijerph-19-12180-f004:**
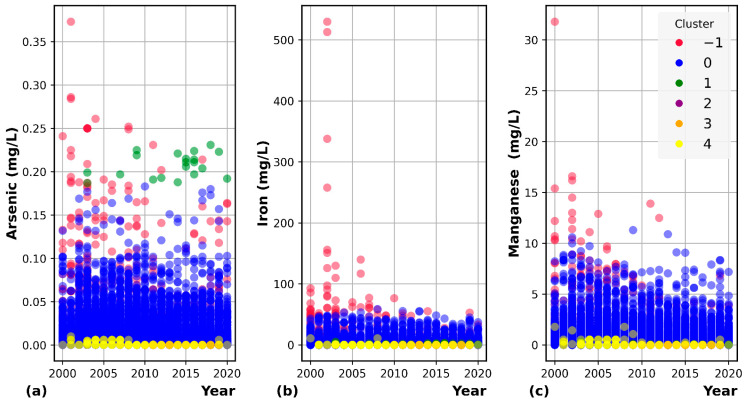
Scatter plot showing the temporal distribution of arsenic (**a**), iron (**b**), and manganese (**c**) in each cluster.

**Figure 5 ijerph-19-12180-f005:**
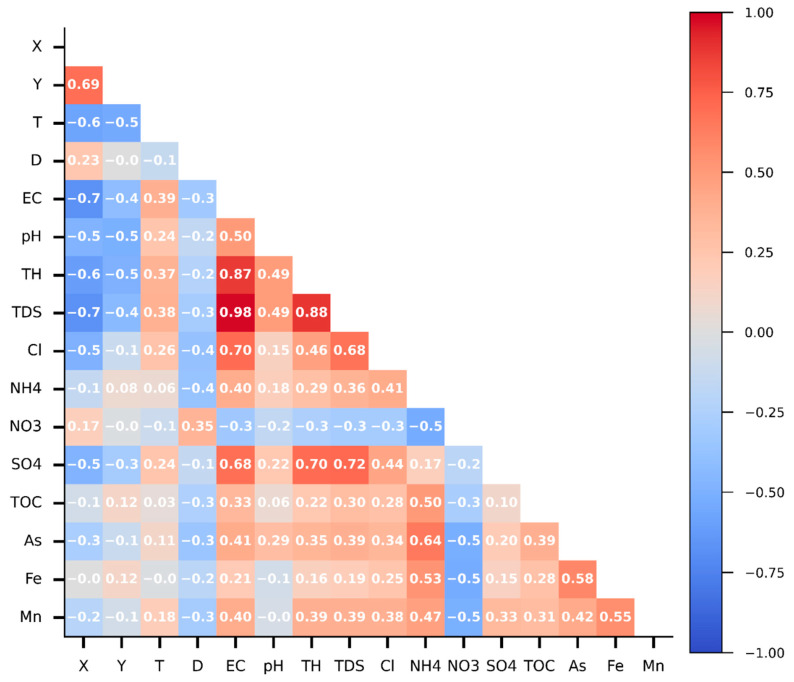
Spearman correlation matrix for all parameters.

**Figure 6 ijerph-19-12180-f006:**
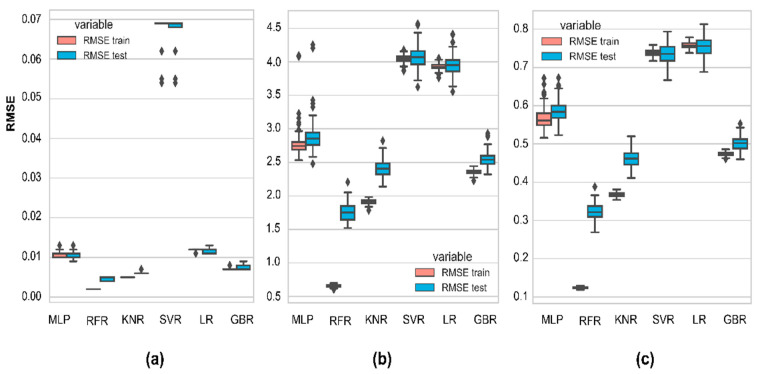
Boxplots of model performance (RMSE) on 100 datasets: (**a**) As prediction; (**b**) Fe prediction; (**c**) Mn prediction.

**Figure 7 ijerph-19-12180-f007:**
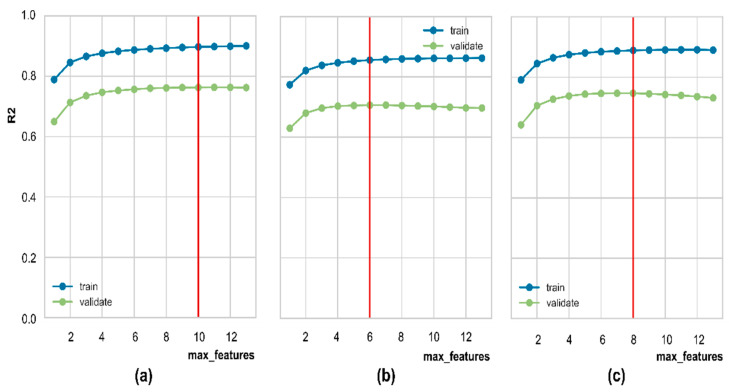
Validation curve of R^2^ score versus max_features: (**a**) As model; (**b**) Fe model; (**c**) Mn model. Vertical red lines indicate optimal max_features.

**Figure 8 ijerph-19-12180-f008:**
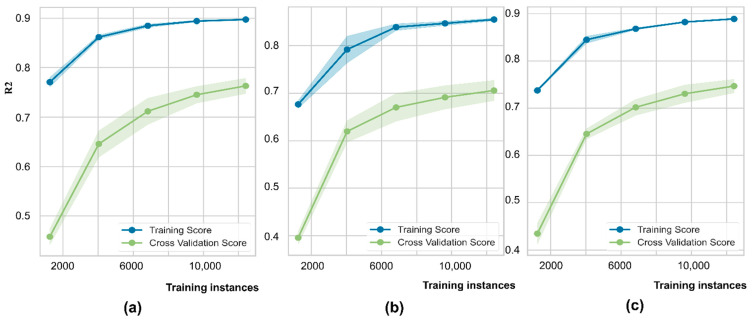
Learning curves of training versus 5-fold cross-validation: (**a**) As model; (**b**) Fe model; and (**c**) Mn model.

**Figure 9 ijerph-19-12180-f009:**
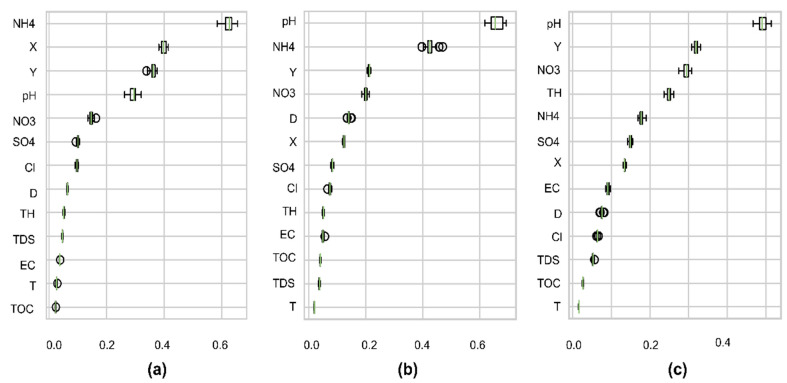
Permute importance distributions on training data: the vertical axis is the feature names, the horizontal axis is the prediction of score reduction when permuting that feature: (**a**) As model; (**b**) Fe model; and (**c**) Mn model.

**Figure 10 ijerph-19-12180-f010:**
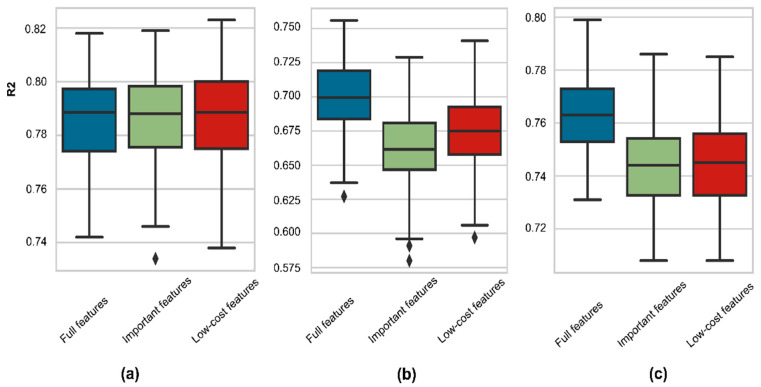
Boxplots of model performance (R^2^) on 100 random testing datasets: (**a**) As model; (**b**) Fe model; (**c**) Mn model.

**Figure 11 ijerph-19-12180-f011:**
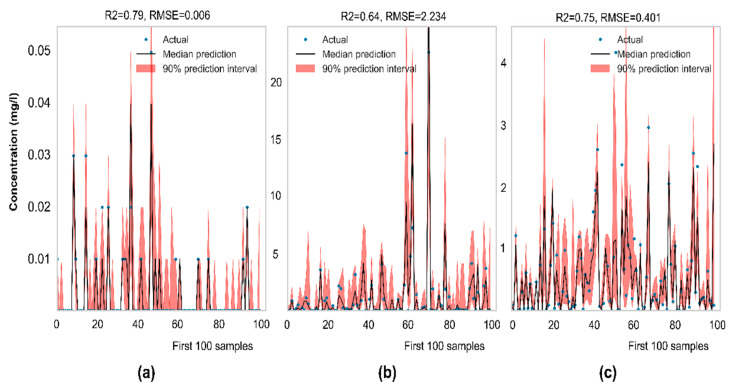
Visualization of observed data, median prediction line, and interval prediction by low-cost features: (**a**) As model, PICP = 97.97%, MPI = 0.015; (**b**) Fe model, PICP = 77.13%, MPI = 4.907; and (**c**) Mn model, PICP = 86.41%, MPI = 1.230.

**Figure 12 ijerph-19-12180-f012:**
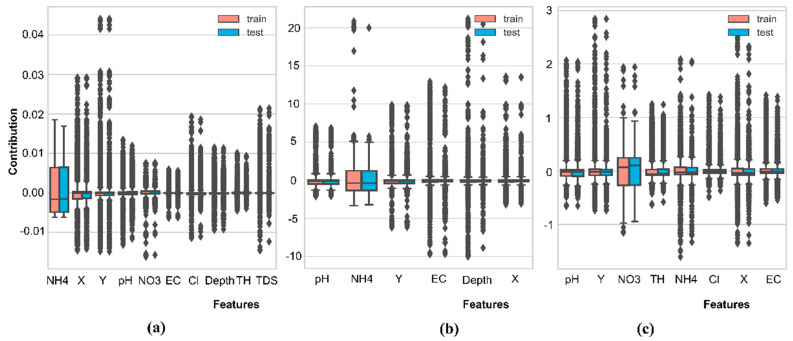
Distribution of local feature contributions for each predicted sample: (**a**) As model; (**b**) Fe model; (**c**) Mn model.

**Table 1 ijerph-19-12180-t001:** Statistical summary of physicochemical parameters.

Variables	Description	Min ^a^	Max ^b^	Mean	STD ^c^	Ske ^d^	Kur ^e^	Unit
Temp	Water temperature	18.600	33.200	26.524	1.704	−0.287	0.449	°C
Depth	Depth to water	0.000	39.627	4.944	4.569	2.777	10.014	m
EC	Electrical conductivity	2.000	65,800.000	1899.373	6205.603	6.419	43.874	μS/cm 25 °C
pH	pH	4.100	9.300	6.739	0.552	−0.922	1.799	-
TH	Total hardness	2.700	8390.000	416.436	735.115	6.207	43.697	mg/L
TDS	Total dissolved solids	4.100	52,300.000	1302.772	4502.321	6.689	48.265	mg/L
Cl	Chloride salt	0.500	27,800.000	454.055	2268.745	6.763	49.321	mg/L
NH_4_	Ammonia Nitrogen	0.001	20.000	0.791	1.704	3.9	19.951	mg/L
NO_3_	Nitrate Nitrogen	0.010	45.500	2.170	3.708	3.22	15.553	mg/L
SO_4_	Sulfate	0.500	4260.000	139.139	312.284	6.386	46.222	mg/L
TOC	Total organic carbon	0.020	15.800	1.922	1.549	2.453	9.442	mg/L
As	Arsenic	0.000	0.146	0.006	0.013	4.031	20.404	mg/L
Mn	Manganese	0.005	11.300	0.519	0.854	4.152	26.062	mg/L
Fe	Iron	0.000	58.800	1.526	4.161	5.821	44.764	mg/L

^a^ minimum; ^b^ maximum; ^c^ standard deviation; ^d^ skewness coefficient; ^e^ kurtosis coefficient.

**Table 2 ijerph-19-12180-t002:** Model performance on 100 testing data sets for different targets (mean R^2^ score ± standard deviation).

Models	As Prediction	Fe Prediction	Mn Prediction
GBR	0.65 ± 0.02	0.59 ± 0.03	0.62 ± 0.02
KNR	0.73 ± 0.02	0.60 ± 0.03	0.65 ± 0.03
LR	0.18 ± 0.01	0.08 ± 0.01	0.18 ± 0.01
MLP	0.30 ± 0.06	0.47 ± 0.06	0.50 + 0.08
RFR	0.79 ± 0.02	0.70 ± 0.03	0.76 ± 0.02
SVR	−26.32 ± 1.88	0.02 ± 0.01	0.21 ± 0.02

**Table 3 ijerph-19-12180-t003:** Random Forest Regressor hyperparameter optimization.

Hyperparameters	Description	As Model	Fe Model	Mn Model
min_samples_leaf	The lowest number of observations in a terminal node	4	4	4
max_features	Number of variables for the best split	10	6	8
min_samples_split	The lowest number of observations needed to split an internal node	6	4	8
n_estimators	Number of trees in a forest	1848	1727	1000
max_depth	The maximum depth of the tree	16	18	20

**Table 4 ijerph-19-12180-t004:** Feature grouping for models.

Feature Set	As Model	Fe Model	Mn Model
Full features	X, Y, pH, TH, EC, TDS, Cl, SO4, NO3, NH4, TOC, Temp, Depth		
Important features	NH4, X, Y, pH, NO3, SO4, Cl, Depth, TH, TDS	pH, NH4, Y, NO3, Depth, X	pH, Y, NO3, TH, NH4, SO4, X, EC
Low-cost features	NH4, X, Y, pH, NO3, EC, Cl, Depth, TH, TDS	pH, NH4, Y, EC, Depth, X	pH, Y, NO3, TH, NH4, Cl, X, EC

**Table 5 ijerph-19-12180-t005:** Results of Wilcoxon-signed rank test on R^2^ scores of 100 testing data sets: statistic (*p*-value).

Paired Tests	As Model	Fe Model	Mn Model
Full features—Important features	2065.000 (0.912)	0.000 (0.000)	0.000 (0.000)
Full features—Low-cost features	1883.000 (0.140)	6.000 (0.000)	0.000 (0.000)
Important features—Low-cost features	1396.500 (0.009)	661.500 (0.000)	1697.000 (0.277)

**Table 6 ijerph-19-12180-t006:** Coverage probability of 90% prediction intervals from different inputs: PICP (MPI).

Feature Sets	As Model	Fe Model	Mn Model
Full features	98.32 (0.0174)	80.53 (6.1914)	87.65 (1.4521)
Important features	98.05 (0.0154)	75.02 (4.8758)	86.31 (1.2356)
Low-cost features	98.07 (0.0152)	77.05 (4.9346)	86.41 (1.2340)

**Table 7 ijerph-19-12180-t007:** Global feature contributions in each model.

Features	As Model		Fe Model		Mn Model	
	Train	Test	Train	Test	Train	Test
Cl	4.33 × 10^−6^	−1.86 × 10^−5^			6.56 × 10^−4^	2.76 × 10^−3^
Depth	4.85 × 10^−6^	3.63 × 10^−5^	4.34 × 10^−3^	−7.28 × 10^−-3^		
EC	4.49 × 10^−6^	−1.71 × 10^−6^	3.19 × 10^−3^	−3.52 × 10^−2^	4.19 × 10^−4^	−2.94 × 10^−3^
NH4	6.13 × 10^−6^	9.45 × 10^−5^	2.56 × 10^−3^	−6.71 × 10^−4^	3.62 × 10^−4^	−1.85 × 10^−3^
NO3	6.87 × 10^−6^	4.68 × 10^−5^			3.64 × 10^−4^	7.00 × 10^−3^
pH	6.86 × 10^−8^	8.14 × 10^−5^	1.22 × 10^−3^	−5.52 × 10^−3^	−2.52 × 10^−4^	−4.82 × 10^−3^
TDS	1.49 × 10^−6^	2.25 × 10^−5^				
TH	3.10 × 10^−6^	6.76 × 10^−6^			5.27 × 10^−4^	4.19 × 10^−3^
X	7.58 × 10^−7^	2.95 × 10^−5^	1.17 × 10^−3^	−4.67 × 10^−3^	2.76 × 10^−4^	−2.61 × 10^−3^
Y	−2.42 × 10^−6^	7.36 × 10^−5^	−2.55 × 10^−4^	−9.63 × 10^−3^	7.97 × 10^−5^	−5.02 × 10^−3^

## Data Availability

Available on Github: https://github.com/trangminhhuynh/GWQ-prediction (accessed on 18 September 2022).

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
