# Peer review of "Predicting Heavy Metal Concentrations in Shallow Aquifer Systems Based on Low-Cost Physiochemical Parameters Using Machine Learning Techniques"

_ijerph, 2022, doi:10.3390/ijerph191912180_

Round 1
Reviewer 1 Report
The issue of ecology at the present stage of society's development is a very important element that guarantees the correct direction of human activity. Ignoring ecological issues is currently unacceptable, especially in the context of the global water deficit in the world. Rivers play an important role in the global hydrological cycle, being an essential element of the ecosystem as well as an important element of the economy and life. For this reason, the quality of water in rivers has been and is of great interest. The existence of man in the natural environment requires constant observation and non-destructive sustainable management, as well as impact in the form of control ensuring minimization of river pollution. The concentration of pollutants transported by rivers is subject to changes in time and space as a result of biological, chemical and physical processes. In water quality supervision systems, mathematical modeling methods can be used, which, in addition to direct measurements, provide the necessary data used in the analysis and interpretation of water quality assessment.
This paper proposes a reliable and explainable framework to find an effective model and feature set to predict heavy metals in groundwater. Early prediction of heavy metal concentration in water can significantly affect the effectiveness of water quality monitoring in ground systems. The random forest algorithm was used, which is one of the most accurate classification methods, consisting in combining multiple classification trees.
Generally the article is written in the correct architecture, but you can pay attention to:
- correct fit of the figures;
- in the conclusions of the work, you can refer to other strategies/methods more broadly;
- making the application available on the GitHub portal, where you can present programming projects for testing.
Author Response
Response to Reviewer 1 Comments
We thank reviewer 1 for the valuable comments. The manuscript has been revised based on the comments and suggestions raised by the reviewer. In the following, we have listed the reviewer's point-by-point comments and the associated responses. Thanks for the valuable comments.
Point 1: correct fit of the figures
Response 1: We have corrected the fit of all figures (please see pages 7, 9-17). Thanks for the comment.
Point 2: in the conclusions of the work, you can refer to other strategies/methods more broadly;
Response 2: We have improved the conclusion as a suggestion. Please see the uploaded manuscript on page 18 for the revision.
Point 3: making the application available on the GitHub portal, where you can present programming projects for testing
Response 3: Thanks for the suggestion. We have created a project on Github and are in the process of modifying our code for users. The input data were uploaded, and the modified code will be ready soon. The relevant information was added at the end of the manuscript in the section “Data Availability Statement: .” Here is the link: https://github.com/trangminhhuynh/GWQ-prediction

Reviewer 2 Report
In this work, the authors apply and compare several machine-learning models for the prediction of 3 heavy metals in groundwater (i.e., As, Fe, and Mn) based on different low-cost physicochemical water-quality indicators (such as NH4, pH, Cl, water-depth, water-temperature, total hardness, etc.). Please consider addressing some major and minor suggestions encountered while reading this interesting work:
1) The authors mention the bootstrap method as one that has some advantages over other methods for data-generation. Although this is true, please consider mentioning that one limitation is at present of long-range dependence (or else known as the Hurst phenomenon (Hurst, 1951), where a strong autocorrelation exists for many lags (above decades), and so, any bootstrap method could destroy this large-lag correlation. This has been identified in other groundwater-quality processes (such the ones analyzed by the authors) by, for example, Dwivedi and Mohanty (2016). Since the Hurst phenomenon is shown to exist in the hydrologial-cycle processes (see a recent review and global-scale analysis in Dimitriadis et al., 2021), it is expected to affect any other water-cycle process, like the ones examined by the authors. Therefore, it is important that this issue is discussed (and, if possible, the authors can also perform some quick analysis to estimate the Hurst parameter; if interested, please do not use the autocovariance-method, since it presents several drawbacks, but rather the climacogram-method, which is shown to outperfom the most common ones in literature).
2) Please consider enriching Table 1 with more statistical metrics of the physicochemical parameters, such as skewness and kurtosis coeffients. Also, please consider including all stations in Table 1 and not just the summary; this could be very helpful to the readers, since they will be able to visualize the variability of the timeseries and reference your work.
3) Please explain why the authors use the Spearman correlation coefficient instead of the most common one by Pearson (see also a recent work by Koutsoyiannis, 2019; where a most accurate correlation coefficient is presented, which is based on the variance instead of the covariance).
4) Please consider explaining whether the authors have include the case of spatial correlation at the spatial characteristics discussed in the Conclusions. For example, when it is mentioned that for the Nutrients a spatial correlation seems to apply, have the authors taken into account spatial correlation between each site or they just present some spatial patterns? In other words, if spatial correlation exist (as for example, it exists between meteorological stations that are very close to each other), then this has to be included in the estimation of the correlation (in the above example, an extreme storm event should not be accounted twice for both stations if they are very close to each other).
5) Please perform a strong grammar and syntax check throughout the document. For example, the sentence "The results show that Random Forest is the most suitable model, and quick-measure parameters can predict arsenic (As), iron (Fe), and manganese (Mn) with promising results but likely uncertainty.", should be changed to something like "The results show that Random Forest is the most suitable model, and when appying the quick-measure parameters, the model can predict arsenic (As), iron (Fe), and manganese (Mn) with promising results but likely with large variability/uncertainty.".
References
Dimitriadis, P., D. Koutsoyiannis, T. Iliopoulou, and P. Papanicolaou, A global-scale investigation of stochastic similarities in marginal distribution and dependence structure of key hydrological-cycle processes, Hydrology, 8 (2), 59, doi:10.3390/hydrology8020059, 2021.
Dwivedi D., and B.P. Mohanty, Hot Spots and Persistence of Nitrate in Aquifers Across Scales, Entropy, 18(1):25, https://doi.org/10.3390/e18010025, 2016.
Hurst, H.E., Long-Term Storage Capacity of Reservoirs, Trans. Am. Soc. Civ. Eng., 116, 770–799, 1951.
Koutsoyiannis, D., Knowable moments for high-order stochastic characterization and modelling of hydrological processes, Hydrological Sciences Journal, 64 (1), 19–33, doi:10.1080/02626667.2018.1556794, 2019.
Author Response
Response to Reviewer 2 Comments
We thank reviewer 2 for the valuable comments. The manuscript has been revised based on the comments and suggestions raised by the reviewer. In the following, we have listed the reviewer's point-by-point comments and the associated responses. Thanks for the valuable comments.
Point 1: The authors mention the bootstrap method as one that has some advantages over other methods for data generation. Although this is true, please consider mentioning that one limitation is at present of long-range dependence (or else known as the Hurst phenomenon (Hurst, 1951), where a strong autocorrelation exists for many lags (above decades), and so, any bootstrap method could destroy this large-lag correlation. This has been identified in other groundwater-quality processes (such as the ones analyzed by the authors) by, for example, Dwivedi and Mohanty (2016). Since the Hurst phenomenon is shown to exist in the hydrological-cycle processes (see a recent review and global-scale analysis in Dimitriadis et al., 2021), it is x expected to affect any other water-cycle process, like the ones examined by the authors. Therefore, it is important that this issue is discussed (and, if possible, the authors can also perform some quick analysis to estimate the Hurst parameter; if interested, please do not use the autocovariance-method, since it presents several drawbacks, but rather the climacogram-method, which is shown to outperform the most common ones in literature).
Response 1: Thank you for your suggestion. We have read the suggested articles and discussed the issues relevant to the Hurst parameter. The discussion was added in the discussion of the manuscript (please see pages 17, 18).
Point 2: Please consider enriching Table 1 with more statistical metrics of the physicochemical parameters, such as skewness and kurtosis coefficients. Also, please consider including all stations in Table 1 and not just the summary; this could be very helpful to the readers since they will be able to visualize the variability of the time series and reference your work.
Response 2: We have improved the descriptive summary in table 1 (please see page 7). Concerning the data of all stations, it would be so distracting to show the data of 403 wells with 20685 samples in the table. The possible option would be to upload data and code on Github for interested researchers. We have created a project on Github and are modifying code files. Here is the link: https://github.com/trangminhhuynh/GWQ-prediction
Point 3: Please explain why the authors use the Spearman correlation coefficient instead of the most common one by Pearson (see also recent work by Koutsoyiannis, 2019; where a most accurate correlation coefficient is presented, which is based on the variance instead of the covariance).
Response 3: In order to capture the relative trend between two variables, we use Spearman correlation instead of Pearson correlation because the Pearson coefficient is used for a linear relationship. In case the data is not exactly linear, using Pearson's coefficient might miss the information that Spearman could capture.
Point 4: Please consider explaining whether the authors have included the case of spatial correlation at the spatial characteristics discussed in the Conclusions. For example, when it is mentioned that for the Nutrients a spatial correlation seems to apply, have the authors taken into account spatial correlation between each site or do they just present some spatial patterns? In other words, if spatial correlation exists (as for example, it exists between meteorological stations that are very close to each other), then this has to be included in the estimation of the correlation (in the above example, an extreme storm event should not be accounted twice for both stations if they are very close to each other).
Response 4: We have examined the predictability of one model for all data, but the results showed that spatial features, like longitude and latitude, have a strong effect on the result. Therefore, we suggested that heterogenous data can be divided into smaller groups in order to capture the local relationship of the data for better prediction (please see page 18).
Point 5: Please perform a strong grammar and syntax check throughout the document. For example, the sentence "The results show that Random Forest is the most suitable model, and quick-measure parameters can predict arsenic (As), iron (Fe), and manganese (Mn) with promising results but likely uncertainty.", should be changed to something like "The results show that Random Forest is the most suitable model, and when appying the quick-measure parameters, the model can predict arsenic (As), iron (Fe), and manganese (Mn) with promising results but likely with large variability/uncertainty.".
Response 5: Thank you so much for your suggestion. We have checked the writing.
